# An Innovative Arteriovenous (AV) Loop Breast Cancer Model Tailored for Cancer Research

**DOI:** 10.3390/bioengineering9070280

**Published:** 2022-06-27

**Authors:** Ran An, Pamela L. Strissel, Majida Al-Abboodi, Jan W. Robering, Reakasame Supachai, Markus Eckstein, Ajay Peddi, Theresa Hauck, Tobias Bäuerle, Aldo R. Boccaccini, Almoatazbellah Youssef, Jiaming Sun, Reiner Strick, Raymund E. Horch, Anja M. Boos, Annika Kengelbach-Weigand

**Affiliations:** 1Laboratory for Tissue Engineering and Regenerative Medicine, Department of Plastic and Hand Surgery, University Hospital Erlangen, 91054 Erlangen, Germany; an_ran2018@163.com (R.A.); majida.al-abboodi@fau.de (M.A.-A.); jrobering@ukaachen.de (J.W.R.); ajay.pharma20@gmail.com (A.P.); theresa.hauck@uk-erlangen.de (T.H.); raymund.horch@uk-erlangen.de (R.E.H.); aboos@ukaachen.de (A.M.B.); 2Department of Plastic Surgery, Union Hospital, Tongji Medical College, Huazhong University of Science and Technology, Wuhan 430022, China; sunjm1592@sina.com; 3Department of Obstetrics and Gynecology, University Hospital Erlangen, 91054 Erlangen, Germany; strisspa@outlook.com (P.L.S.); reiner.strick@uk-erlangen.de (R.S.); 4Institute of Genetic Engineering and Biotechnology, University of Baghdad, Baghdad 10081, Iraq; 5Department of Plastic- and Hand Surgery, University Hospital RWTH Aachen, 52074 Aachen, Germany; 6Institute of Biomaterials, Friedrich-Alexander University Erlangen-Nürnberg, 91056 Erlangen, Germany; supachai.reakasame@gmail.com (R.S.); aldo.boccaccini@ww.uni-erlangen.de (A.R.B.); 7Institute of Pathology, University Hospital Erlangen, 91054 Erlangen, Germany; markus.eckstein@uk-erlangen.de; 8Institute of Clinical Radiology, University Hospital Münster, 48149 Münster, Germany; 9Preclinical Imaging Platform Erlangen (PIPE), Department of Radiology, University Hospital Erlangen, 91054 Erlangen, Germany; tobias.baeuerle@uk-erlangen.de; 10Department of Functional Materials in Medicine and Dentistry, University of Würzburg, 97080 Würzburg, Germany; moataz.youssef@fmz.uni-wuerzburg.de; 11Institute of Pathology, University of Würzburg, 97080 Würzburg, Germany

**Keywords:** arteriovenous loop, breast cancer, animal model

## Abstract

Animal models are important tools to investigate the pathogenesis and develop treatment strategies for breast cancer in humans. In this study, we developed a new three-dimensional in vivo arteriovenous loop model of human breast cancer with the aid of biodegradable materials, including fibrin, alginate, and polycaprolactone. We examined the in vivo effects of various matrices on the growth of breast cancer cells by imaging and immunohistochemistry evaluation. Our findings clearly demonstrate that vascularized breast cancer microtissues could be engineered and recapitulate the in vivo situation and tumor-stromal interaction within an isolated environment in an in vivo organism. Alginate–fibrin hybrid matrices were considered as a highly powerful material for breast tumor engineering based on its stability and biocompatibility. We propose that the novel tumor model may not only serve as an invaluable platform for analyzing and understanding the molecular mechanisms and pattern of oncologic diseases, but also be tailored for individual therapy via transplantation of breast cancer patient-derived tumors.

## 1. Introduction

Breast cancer is the leading cause of female mortality and morbidity in the world [1]. The major challenge in advancing our understanding of the biology of breast cancer is the availability of experimental models that can recapitulate breast cancer carcinogenesis and progression [2]. In order to bridge the gap between preclinical and clinical studies, numerous efforts have been made to develop models of breast cancer [3,4]. Using in vitro systems, e.g., two-dimensional (2D) or three-dimensional (3D) cultivation of cancer cells in different matrices or scaffolds, it is difficult to simulate the complex tumor–stromal interactions that contribute to key aspects of breast cancer biology such as the angiogenic switch [5]. Breast cancer is heavily influenced by the tumor microenvironment (TME), making in vivo models desirable to mimic the heterogeneity of breast cancer as accurately as possible to study unknown molecular processes and find new therapeutics [6]. For these reasons, in vivo models remain critically important for investigating breast cancer.

An ideal animal model of human cancer would reproduce the genetic and phenotypic changes occurring in human cancer [7]. Vascularization is essential, as engineered tumors require a functional blood vessel network to supply the nutrients and oxygen for their survival at the implantation site. The most common animal model for solid tumors is the subcutaneous tumor xenograft, in which the surrounding tissue may play a decisive role in altering the behavior and metastatic potential of the engrafted tumor cells [2,6]. Hence, some more advanced animal models, such as orthotopic models, which can provide a more favorable microenvironment, have been developed to overcome these shortcomings [2].

The arteriovenous (AV) loop animal model, formed by microsurgical anastomosis of a vein and artery, can create an isolated microenvironment in vivo within a closed implantation chamber. Within this isolated environment, which is only connected to the living organism by the AV loop vessels, the tumor can grow independently of the surrounding tissue, facilitating the analysis of the developing tumor without disturbing influences. Originally, the AV loop model was developed for engineering vascularized tissues. By varying the matrices and cells in the implantation chamber, a spectrum of different tailored tissues to a particular research area can be generated [8]. In our previous study, we successfully established a melanoma model in alginate/hyaluronic acid/gelatin hydrogel in the AV loop [9]. However, to the best of our knowledge, the AV loop model has never been employed to generate breast tumors within the isolated chamber.

Because direct cell implantation procedures into animals remarkably lead to a loss of living cells with survival rates as low as 1–32%, it is necessary to identify a biocompatible matrix that can be further biochemically and physically modified to maximize cell retention rates and promote cell–cell and cell–matrix interactions [10,11]. Hydrogels, obtained from either natural or synthetic polymers, are candidates to facilitate tumor formation in a 3D environment. Hydrogels are similar to the extracellular matrix (ECM) in terms of porosity, flexibility, and high water preservation, leading to enhanced permeability of oxygen and reduced mechanical damage to tissues [12,13,14,15]. Biologically derived matrices, such as fibrinogen, facilitate cell adhesion and cell signaling due to their cellular binding domains [15]. However, in some cases, the poor mechanical properties of natural hydrogels have imposed significant limitations on their use in tissue regeneration [15]. In contrast, synthetic polymers can be fabricated in a reproducible manner, allow better control because of their mechanical properties, and have negligible immunogenicity [16]. Porous scaffolds made from poly(lactide-co-glycolide) PLGA or chitosan have been used to engineer tumor spheroids with angiogenic characteristics and malignant potential [17,18]. Polycaprolactone (PCL), a polymer with long-term biodegradability, is already in clinical use in several medical devices and also provides a stable 3D culture substrate applicable for in vitro purposes [19]. PCL scaffolds are increasingly used due to their ease of processing, handling capability, and access to medical-grade raw material [20].

This study aimed to develop a human breast cancer model using the AV loop rat model by implantation of breast carcinoma cells (HTB-26) into the biomaterials fibrin, alginate, and a mixture of fibrin and alginate with PCL scaffolds into the AV loop chamber. Tumors were analyzed by in vivo 3D imaging after 4 and 8 weeks and immunohistochemistry after 8 weeks. We propose that this tumor model provides an adaptable microenvironment within a living organism where tumor progression and especially vascularization can be analyzed without the influence of the surrounding rat tissue. We propose that this AV loop in vivo tumor model may find broad utility in cancer research where the microenvironment can be controlled.

## 2. Materials and Methods

### 2.1. HTB-26 Cell Culture

Human breast cancer cell line HTB-26 (MDA-MB-231) was purchased from American Type Culture Collection (ATCC, Rockville, MD, USA). Cells were cultured in Dulbecco’s Modified Eagle Medium DMEM (Biochrom AG, Berlin, Germany) supplemented with 10% fetal calf serum (FCS) (FCS superior, Biochrom AG, Berlin, Germany), 1% L-glutamine (Sigma-Aldrich, St. Louis, MO, USA) and 1% Non-Essential Amino Acid Solution (NEAA) (Thermo Fisher Scientific Inc., Waltham, MA, USA) in a humidified incubator under an atmosphere of 95% air and 5% CO_2_ at 37 °C. Cell passaging was performed every 3 or 4 days when the monolayer of adherent cells reached 80–90% confluence.

### 2.2. Preparation of Polycaprolactone Scaffolds

Polycaprolactone (PCL) scaffolds were fabricated using melt electrowriting. A 20-layer-PCL scaffold with 200 µm × 200 µm pores was fabricated following methodologies as previously described [21]. Briefly, 40 mm × 40 mm scaffolds were printed and then laser cut into 8 mm circles using a CO_2_ laser cutter (Rayjet 50 C30, Trotec Laser GmbH, Marchtrenk, Austria). The PCL scaffolds were immersed in 100% ethanol for 5 min to remove air bubbles, then incubated in sodium hydroxide (1 μM) for 30 min at room temperature, followed by two washes with phosphate-buffered saline (PBS). Prior to cell culture experiments and implantation, the scaffolds were incubated with 10 µg/mL human fibronectin (Thermo Fisher Scientific Inc., Waltham, MA, USA) for 1 h at room temperature. Next, 3 × 10^5^ HTB-26 were seeded on each PCL scaffold placed into 0.5% alginate (PH175, VIVAPHARM^®^, JRS PHARMA GmbH & Co., KG, Rosenberg, Germany) -coated wells of a 24-well plate containing 500 μL of culture media.

### 2.3. Preparation of Fibrin Gel

Fibrin gels were prepared with a total cell amount of 1.25 × 10^6^ HTB-26/mL. Cells were diluted in different concentrations of fibrinogen in a 24-well plate and polymerized with thrombin (final concentration in hydrogels 10 IU/mL) (all from Baxter GmbH, Vienna, Austria) to final fibrin hydrogel concentrations of 1% (*w*/*v*), 2% (*w*/*v*), 3% (*w*/*v*), or 4% (*w*/*v*). Fibrin gels were incubated for 10 min to allow formation and stabilization of the hydrogels. A quantity of 500 µL cell culture medium was added, and hydrogels were incubated at 37 °C in 5% CO_2_ for up to 18 days.

### 2.4. Cell Encapsulation in Alginate-Fibrin Matrices

To encapsulate cells, 1% (*w*/*v*) fibrinogen (Baxter GmbH) and 0.5% (*w*/*v*) alginate (VIVAPHARM^®^) were mixed with 5 × 10^6^ HTB-26 cells and filled in the syringe of the extruder (Nordson EFD, East Providence, RI, USA), which was connected to a high precision fluid dispenser (Ultimus V; Nordson EFD, Feldkirchen, Germany). The cell droplets were crosslinked in a calcium chloride solution (0.1 M) for 15 min and washed two times in Hanks’ Balanced Salt Solution (HBSS) (Millipore Sigma, Burlington, MA, USA). Subsequently, capsules were incubated in 20 IU/mL thrombin (Baxter GmbH, Vienna, Austria) for further hardening.

### 2.5. Sacrificial Gel Preparation

The sacrificial material used in this study contained 9% (*w*/*v*) methyl cellulose (MC) (4000 cP, Sigma-Aldrich) and 5% (*w*/*v*) gelatin Type A (Bloom 300, porcine skin derived, Sigma-Aldrich). MC powder was sterilized by autoclaving, and 5% (*w*/*v*) gelatin solution was sterilized by filtering through 0.22 µm filters (Carl Roth GmbH & Co., KG, Karlsruhe, Germany). The gelatin solution was mixed with MC powder by stirring for 30 min at 50 °C. The mixtures were incubated at 37 °C for 2 h and stored overnight at 4 °C. Prior to printing, the sacrificial gel was kept at room temperature for 24 h.

### 2.6. Preparation of the Alginate–Fibrin Three-Dimensional Ring-/Round-Shaped Matrices

To prepare the custom-made alginate–fibrin hydrogels for highest fitting accuracy into the in vivo implantation chambers, the sacrificial gel was placed in a cartridge with a conical plastic nozzle (G22, Nordson EFD, Feldkirchen, Germany) with an inner diameter of 410 µm. Subsequently, the sacrificial ink was extruded with 300 KPa air pressure and 5 mm/s printing speed using a 3D printer (BioScaffolder 2.1, GeSiM, Radeberg, Germany). To fabricate ring-shaped alginate–fibrin hydrogels, seven layers of the sacrificial ink were printed in a layer-by-layer fashion to generate two concentric rings with diameters of 4.4 and 10 mm, respectively. Subsequently, the space between the two rings was filled with 70 µL 0.5% alginate and 1% fibrinogen solution containing 1.5 × 10^6^ HTB-26 cells. To fabricate the round-shaped plate with four holes, the sacrificial ink was printed in a layer-by-layer fashion to generate one outer ring with a diameter of 13 mm and four inner rings with a diameter of 1 mm. Next, the space between the rings was filled with 70 µL 0.5% alginate and 1% fibrinogen hybrid matrices. The samples were immersed in 100 mM CaCl_2_ for crosslinking. The alginate-fibrin matrices were carefully detached and then transferred into a culture plate containing 20 IU/mL thrombin for further hardening. Subsequently, the ring-shaped and round-shaped constructs were incubated in DMEM (Biochrom AG, Berlin, Germany) for 1 d under cell culture conditions to dissolve the sacrificial structures. The ring- and round-shaped constructs are shown in Appendix A.

### 2.7. Scanning Electron Microscopy (SEM) Analysis

The melt electrowritten PCL scaffolds were examined by scanning electron microscopy (SEM) (AURIGA CrossBeam; Carl Zeiss Microscopy GmbH, Oberkochen, Germany). Prior to microscopy, the samples were coated with a 7.5 nm gold layer using a sputter coater (Q150T Turbo-pumped Sputter Coater, Quorum Technologies Inc., Guelph, ON, Canada).

### 2.8. Immunofluorescence In Vitro

For wheat germ agglutinin (WGA) immunofluorescence, PCL scaffolds seeded with cells on day 7 of culture were incubated with WGA (2 mg/mL, Sigma-Aldrich), diluted 1:100 in PBS for 10 min at room temperature. The scaffolds with cells were counterstained with Hoechst 33342 (10 mg/mL, Sigma-Aldrich), diluted 1:400 in PBS for 5 min at room temperature to visualize the nuclei.

To identify HTB26 cells on PCL scaffolds, an anti-pan-CK antibody (types 5, 6, 8, 17, and 19) (1:100) (mouse anti-human, clone MNF 116, DAKO, Glostrup, Denmark) and Alexa Fluor^®^ 488 (1:500) (goat anti-mouse, Life Technologies, Carlsbad, CA, USA) were used.

To analyze cell proliferation on PCL scaffolds after 3 days of incubation, the samples were fixed, permeabilized, and blocked, followed by incubation with a primary Ki67 antibody (1:100) (mouse anti-human, clone MIB-1, DAKO) and a secondary antibody Alexa Fluor^®^ 488 (1:200) (goat anti-mouse, Life Technologies, Carlsbad, CA, USA), and the nuclei were counterstained with DAPI (1:1000) (Roche Molecular Systems Inc., Pleasanton, CA, USA).

For quantification of cell proliferation in fibrin hydrogels after 18 days, 10 µm cryosections were cut from three different planes of the hydrogels. Slides were stained with a Ki67 antibody (1:70) (rabbit anti-human, monoclonal SP6, Zytomed Systems GmbH, Berlin, Germany) and the CSA II Biotin-free Tyramide Signal Amplification System (DAKO). Proliferating cells were counted in four different regions of interest (ROI) in 200× magnification, and the percentage of proliferating cells was calculated.

Pictures were taken by fluorescence microscopy (Olympus IX83, CellSens software; Olympus, Tokyo, Japan).

### 2.9. Cell and Hydrogel Preparations for In Vivo Implantation

Prior to implantation, different hydrogels were prepared for each group. For the fibrin group, fibrinogen and thrombin (both from Baxter GmbH) were prepared to achieve final concentrations in the hydrogels of 2% fibrinogen and 10 IU/mL thrombin, respectively. For implantation, 5 × 10^6^ HTB-26 were diluted in 100 µL thrombin. For the alginate-fibrin group, fibrinogen and alginate were mixed to the final concentrations 1% and 0.5%, respectively, and encapsulated with or without 5 × 10^6^ HTB-26 as described above. For the PCL group, each scaffold was seeded with 3 × 10^5^ HTB-26 cells 1 week before implantation as described above for the in vitro experiments. Two alginate–fibrin 3D rings with 1.5 × 10^6^ HTB-26 per ring, two pieces of alginate–fibrin 3D sheets, and alginate–fibrin capsules without cells were prepared 1 d before implantation. For the control group, 1% fibrinogen and 0.5% alginate capsules without any cells were produced 1 d prior to implantation.

### 2.10. Surgical Procedure

To investigate the effects of fibrin, alginate–fibrin, and PCL scaffolds on tumor formation and vascularization, a T-cell-deficient, athymic nude rat model (Crl:NIH-*Foxn*1^rnu^ Rat, male, weight: 300–400 g; Charles River Laboratories, Wilmington, MA, USA) was used and randomly divided into four groups. The groups were as follows: 2% fibrin group with cells (*n* = 5), alginate–fibrin group with cells (*n* = 8), PCL group with cells (*n* = 9), and a control group of alginate–fibrin capsules without cells (*n* = 6). All experiments were carried out in accordance with the Animal Care Committee of the University of Erlangen-Nürnberg and the Government of Unterfranken, Germany (license number 55.2 DMS-2532-2-352). Rats were housed in a standardized IVC-environment (20–22 °C, relative humidity 46–48%, light/dark cycles of 12 h), and they had free access to water and standard chow.

The arteriovenous (AV) loop surgery was performed as described previously [8]. Briefly, rats were anesthetized with isoflurane (Baxter Deutschland GmbH, Unterschleißheim, Germany), and the operation field was prepared to achieve sterility. Anesthesia was maintained by inhalation of 1–2% isoflurane in oxygen. At the inner side of the right thigh, vessels were microsurgically prepared and a 1.5-cm-long femoral vein graft was harvested. The femoral vascular bundle was microsurgically prepared from the groin of the left thigh to the bifurcation of the femoral artery in the knee. After harvesting the venous graft from the right femoral vein, the loop was formed by anastomosing the proximal end of the venous graft with the proximal end of the vein and the distal end of the venous graft with the proximal end of the artery with a 11-0 suture (Ethicon Inc., Somerville, NJ, USA) (Figure 1A). The construct compositions and final cell number in different groups is shown in Appendix A. For the fibrin group, filling of the custom-made Teflon isolation chamber was performed in a 4-step-process. First, a layer of 400 µL fibrin was pipetted into the chamber and allowed to clot. Second, the AV loop was placed onto the first layer and a fibrin clot of 200 µL was placed at the entrance of the Teflon chamber to prevent leakage of tumor cells. Third, 5 × 10^6^ HTB-26 cells, suspended in 100 µL thrombin, were mixed with 100 µL fibrinogen and pipetted around the loop vessels. Forth, the chamber was completely filled with additional 400 µL fibrin and closed with the lid. For the alginate–fibrin group, a layer of alginate–fibrin capsules without cells was placed on the bottom of the chamber. Next, the AV loop was placed onto the first layer. Subsequently, 5 × 10^6^ HTB-26 cells encapsulated in 200 µL alginate–fibrin were put around the AV loop. Finally, alginate–fibrin capsules without cells were pipetted into the chamber for complete filling. For the PCL group, a piece of 3D printed alginate–fibrin sheet without cells was placed into the chamber. Second, three PCL scaffolds with HTB-26 cells were put onto the alginate–fibrin sheet. Third, the AV loop was put on the scaffolds. Forth, one 3D alginate–fibrin ring with cells was placed inside the loop (Figure 1B) and the other ring with cells was put around the AV loop. Fifth, another three PCL scaffolds with HTB-26 cells were put onto the AV loop (Figure 1C). Sixth, one more piece of 3D printed alginate–fibrin sheet without cells was placed on the top, and everything was fixed with 2% fibrin (Figure 1D). The chamber was fixed at the thigh with sutures (Figure 1E) and the skin closed. For the control group, first, a layer of alginate–fibrin capsules without cells was placed on the bottom of the chamber. Next, the AV loop was put on the capsules. Subsequently, capsules without cells were used for filling the chamber. For the microsurgeries, an operating microscope (Leica M655, Wetzlar, Germany) was used. Postoperatively, animals were given 7.5 mg/kg enrofloxacin (Bayer AG, Leverkusen, Germany) subcutaneously (s. c.) to prevent infections and the analgesic tramadol (Grünenthal GmbH, Aachen, Germany) in a dosage of 12.5 mg/kg per os (p.o.) for 5 days. Further, 1 mg/kg/day prasugrel (Daiichi Sankyo GmbH, Munich, Germany) p.o. was used once for 3 d, and 10 mg/kg of low molecular weight heparin (Enoxaparin sodium; Sanofi-Aventis GmbH, Frankfurt am Main, Germany) (s. c.) was applied twice daily during one week after implantation.

### 2.11. In Vivo and In Vitro Imaging

In vivo imaging was performed at weeks 4 and 8 after tumor cell inoculation using MRI (for fibrin, alginate–fibrin and control group, *n* = 2; for PCL group, *n* = 4) and PET-CT (for fibrin and alginate–fibrin group, *n* = 2; for PCL group, *n* = 4) under general anesthesia with isoflurane. Due to the small animal number in imaging, no statistical comparison between the groups was performed. Further imaging was performed after explantation using micro-CT (for fibrin group, *n* = 1).

### 2.12. MRI

MRI was performed on a preclinical 7T ultra high field scanner (ClinScan 70/30, Bruker, Ettlingen, Germany) using a dedicated volume resonator radiofrequency coil (Bruker). Dynamic contrast-enhanced MRI (DCE-MRI) was performed using a fast low-angle shot (FLASH) sequence with the following parameters: repetition time (TR)/echo time (TE): 3.9/0.88 ms, field of view (FOV): 65 × 95, matrix: 256 × 256, in-plane resolution (res): 0.254 mm × 0.254 mm, slice thickness: 1.0 mm, averages (av): 1, measurements: 100, acquisition time (TA): 9:10 min. After 30 s baseline, 0.3 mmol/kg Gd-DTPA (Magnevist, Schering, Germany) was infused intravenously over a time period of 10 s via a tail vein catheter. Before and after DCE-MRI, a morphologic T1-weighted spin echo sequence (TR/TE: 400/6.7 ms, FOV: 65 × 95, matrix: 320 × 320, res: 0.2 mm × 0.2 mm, slice thickness: 0.7 mm, av: 2, TA: 4:08 min) was applied.

### 2.13. PET-CT

PET-CT was performed on a preclinical hybrid scanner (Inveon, Siemens, Erlangen, Germany). A quantity of 5.5 MBq of 2-deoxy-2-[fluorine-18]fluoro-D-glucose (18F-FDG) was intravenously injected into the animals 25 min prior to imaging. CT imaging was performed with the following settings: tube voltage: 80 kV, tube current: 500 µA, acquisition: step-and-shoot, rotation: full, settle time: 500 ms, projections: 180, exposure time: 1100 ms, binning: 2 × 2, charge-coupled device (CCD) size: 2688 × 2048 px, FoV: 66 mm × 50 mm, effective pixel size: 49 µm, scan time: 6 min. After CT, PET images were acquired for 15 min. PET data were analyzed using Inveon Acquisition software (Siemens). For each animal, mean and maximum activities in the skin chamber were determined. The parameter area under the curve (AUC, associated with blood volume) was derived using a self-written software script, and mean standard uptake values (SUV mean) were calculated by dividing the obtained mean activities by the injected activity and multiplying the result by the weight of the individual animal.

### 2.14. Explantation Procedure

Eight weeks after the AV loop implantation, animals were anesthetized with isoflurane and the vascular system was perfused with the radiopaque agent Microfil^®^ MV-122 (Flowtech Inc., Carver, MA, USA). The abdominal aorta was cannulated and V. cava inferior opened. Subsequently, the vascular system was first flushed with warm heparinized (100 I.E/mL) physiological sodium chloride solution (Ratiopharm GmbH, Ulm, Germany) until there was clear backflow and then flushed with 30 mL yellow Microfil^®^ MV-122 containing 5% of MV Curing Agent (both from FlowTech Inc.). The perfused rats were stored at 4 °C for 24 h for hardening of the Microfil^®^ solution. Subsequently, constructs were explanted and fixed in formalin for 24 h at 4 °C.

### 2.15. Micro-CT Analysis

Micro-CT was performed after explantation (for fibrin group, *n* = 1). The constructs were scanned in a Micro Computed Tomograph (Inveon, Siemens, Erlangen, Germany) with a voltage of 80 kV, a tube current of 500 µA, and an exposure time of 1100 ms per rotation step. An effective isotropic resolution of 50 µm was reached. Image post processing was performed with the medical image viewer OsiriX (Pixmeo SARL, Bernex, Switzerland).

### 2.16. Histological Analysis

From each paraffin-embedded explant, 3-µm-thick sections were cut. For quantification of the explanted construct size, six hematoxylin and eosin (H&E) stained sections from different planes were completely scanned at 40× magnification (Olympus IX83, cellSens Software). The analysis of the area of the cross section was performed semi-automatically (Leica Application Suite V3, Leica Microsystems Wetzlar, Germany). Mean values of the planes were included for statistical analysis.

### 2.17. Immunohistochemically Staining

A rabbit anti-rat CD31 antibody (1:70, clone N/A) and the ZytoChem-Plus AP Polymer-Kit (all from Zytomed Systems GmbH) were used to quantify newly formed vessels. A citrate buffer with pH 6 was applied as antigen retrieval. On 2–3 different slides of each explant, the number and area of CD31-positive vessels per construct area were counted and measured in 1–4 ROIs (depending on the size of the construct) per slide at 200× magnification. For each explant, 5–8 ROIs in total were measured. Mean values were included for statistical analysis. 

Cell proliferation was determined by using a rabbit anti-Ki67 antibody (1:200, clone SP6; Zytomed Systems GmbH) and the CSA II Biotin-free Tyramide Signal Amplification System (DAKO) after pretreatment with a citrate buffer of pH 6. For quantification of the cell viability, 4 ROIs at 200× magnification per slide were analyzed. Two slides from different planes of the explants were used for quantification.

Apoptosis was detected by the FragEL^™^ DNA Fragmentation Detection Kit (TUNEL assay, Calbiochem^®^, Merck KGaA, Darmstadt, Germany). Counterstaining was performed with DAPI (1:1000) (Roche Molecular Systems Inc.).

For visualization of cells of the monocyte/macrophage lineage, a mouse anti-rat CD163 (1:300, clone ED2, Bio-Rad Laboratories, Hercules, California, USA) and a mouse anti-rat CD68 (1:100, clone ED1, AbD Serotec, Kidlington, UK) were used in combination with the ZytoChem-Plus AP Polymer-Kit after antigen retrieval with pronase (Millipore Sigma). Further, to detect CD8-positive cells, a mouse anti-rat CD8 alpha antibody (clone OX-8, Abcam, Cambridge, UK) was used in a concentration of 1:200 with the HRP anti rabbit/anti mouse Envision Kit (DAKO). A citrate buffer of pH 6 was applied as antigen retrieval.

To detect tumor cells and growing tumor tissue within the explants, a mouse pan-cytokeratin (CK) anti-human/mouse/rat/dog antibody (CK1-6, 8, 10, 14-16, 19, Zytomed Systems GmbH) was used in a concentration of 1:50 in combination with the POLAP-100 AP Polymer System (Zytomed Systems GmbH). For vimentin staining, a mouse anti-vimentin anti-human/rat/mouse/dog/pig antibody (clone V9 Zytomed Systems GmbH) was used in a concentration of 1:150 in combination with HRP anti rabbit/anti mouse Envision Kit (DAKO).

Anti-human PD-L1 and CD3 immunohistochemistry was performed using an automated Ventana Benchmark Ultra autostainer (Ventana, Tucson, AZ, USA). PD-L1 detection was performed using a commercially available assay kit (SP263 assay, Ventana, USA) and CD3 at a dilution of 1:50 (Clone F7.2.38, monoclonal mouse, ThermoFisher Scientific Inc., Waltham, MA, USA). CD3 staining was further performed on rat blood cells to show cross-reactivity with rat cells.

### 2.18. Statistical Evaluation

Data are expressed as mean ± standard deviation. Statistical analysis was performed with SPSS 21.0 for Windows (SPSS Inc., Chicago, IL, USA). Results were interpreted statistically using the non-parametric Kruskal–Wallis test and the Mann–Whitney U-Test. The level of statistical significance was set at *p* ≤ 0.05. All graphics were created with GraphPad 8.0. Figures were created with CorelDRAW X6. The brightness, contrast, and intensity of the depicted images were adapted for better perceptibility. Adaptations were made to the entire picture.

## 3. Results

### 3.1. Successful Growth of Breast Cancer Cells on Scaffolds and Matrices

The PCL-scaffolds were printed using melt electrowriting (MEW) and exhibited a box pore-shaped homogenous morphology (Figure 2A). Using WGA staining and cytokeratin immunofluorescence staining (Figure 2B,C), it was confirmed that HTB-26 cells successfully attached on the PCL scaffolds and proliferated along the pore walls. The vast majority of living cells on PCL scaffolds stained Ki67-positive (Figure 2D), indicating favorable proliferative activity on day 3 of culture.

Immunofluorescence results for Ki67 of HTB-26 cells cultured in different concentrated fibrin matrices showed that cells proliferated significantly better in 1% (*w*/*v*) and 2% (*w*/*v*) fibrin compared to 4% (*w*/*v*) fibrin on day 18 of culture (*p* < 0.01) (Figure 2E). No significant difference was observed among 1% (*w*/*v*), 2% (*w*/*v*), and 3% (*w*/*v*) groups. In fibrin matrices, cells were homogenously distributed and small colonies appeared after 4 days of culture (Figure 2F,G). Additionally, HTB-26 cells were encapsulated in an alginate–fibrin matrix. Figure 2H,I show representative phase contrast images for encapsulated HTB-26 cells at 1 and 4 d after inoculation. At 1 d after encapsulation, the capsules were intact and retained their round structure. No visible degradation was observed. On day 4, some cells escaped from the capsules, attached at the bottom of the cell culture plate, and exhibited normal morphology (Figure 2I). To facilitate the implantation, the alginate–fibrin matrices were 3D-produced in a ring shape with an outer diameter of approximately 10 mm (Figure 2J).

### 3.2. In Vivo and In Vitro Imaging Confirms Tumor Formation and Vascularization

For characterization of newly formed tumor tissue with vascularization, in vivo MRI and PET-CT was performed. In particular, T1-weighted sequences with contrast enhancement provided information on perfused areas within the implantation chamber. In all groups, the growing vasculature could be visualized over time (Figure 3A–D). Tumor formation was visible as tracer-avid areas after 4 weeks in PET-CT imaging (Figure 3B–D). All rats that were imaged and implanted with HTB-26 cells were scored positively by PET-CT scanning. Increased FDG-uptake contrast in the tumor as compared to surrounding muscle tissue was observed at the time points of 4 weeks and 8 weeks. As shown in Figure 3E,F, area under curve (AUC) and mean standard uptake values (SUV_mean_) in alginate–fibrin and SUV_mean_ in fibrin group increased from 4 weeks to 8 weeks. However, in the PCL and fibrin groups, the value of AUC was decreasing slightly, and no visible change of SUV_mean_ was detected in the PCL group from 4 weeks to 8 weeks.

To further monitor the 3D microvasculature, micro-CT using an intravascular contrast agent was performed after 8 weeks. An image of the fibrin group is shown in Figure 3G. Micro-CT vividly depicts the 3D morphology of the AV loop and the vessel branches generated from the AV loop in fibrin, revealing angiogenesis within the chamber.

### 3.3. Morphometric Analysis of AV Loop Breast Cancer Microtissues

Eight weeks after implantation, the generated microtissues were explanted and prepared for histology. After confirming the patency of the AV loop vessels based on the microscopic observations of the H&E staining, in total *n* = 5 constructs per group were included for the histological evaluation. The microtissue size was quantified by measuring the cross section area microscopically. Particularly, the alginate–fibrin group exhibited a significant larger area compared to the other groups (Figure 4A). The average section area in the alginate–fibrin groups was nearly twice as large as that in the control and PCL groups. No significant difference was observed between the control and PCL groups. In contrast, the microtissue size of the fibrin group was significantly lower than that of the other groups, which was consistent with the macroscopic appearance (Figure 4B,D,F,H).

Notably, macro- and microscopically, the generated microtissues of the control (Figure 4B,C), alginate–fibrin (Figure 4D,E), and PCL (Figure 4F,G) groups retained the most volume compared to the fibrin group (Figure 4H,I), which degraded highly over 8 weeks. A clear demarcation between newly formed human tumor colonies and rat tissue indicated by morphological differences could be observed. All constructs were well-vascularized originating from the AV loop.

### 3.4. Human Breast Cancer Tumor Growth within the AV Loop Chamber In Vivo

Vascularized human tumor tissue could successfully be generated within the AV loop chamber. Immunohistological analyses of the constructs revealed that the origin of the tissue was derived from breast cancer cells, which stained positive for pan-cytokeratin but also for vimentin (Figure 5A–D). A high level of pan-cytokeratin and vimentin expression was observed mainly in the alginate–fibrin group. Large areas of solid tumor formation were visible. In contrast, in PCL groups, there were only small CK-positive tumor colonies. The formation of several tumor nodules was confirmed by CK and vimentin expression of the fibrin group. Interestingly, the positive distribution displayed a clear homogeneity in cytokeratin and vimentin staining. Notably, no positive cells were observed in the control group.

### 3.5. High Proliferation and Low Apoptosis in the AV Loop Constructs

Vast amounts of Ki67-positive cells were detected in all groups 8 weeks after implantation, indicating a high proliferation rate around 32–40% of cells on scaffolds (Figure 6A,B). Diffuse cytoplasmic immunoreactivity for Ki67 was noted in the majority of tumor cells. Conversely, only rare Ki67 nuclear expression was found. No significant differences were detected in the percentage of proliferating cells (Figure 6B). In comparison, a few apoptotic cells with no visible difference using the TUNEL assay were detected in all groups (Figure 6A).

### 3.6. Well-Vascularized AV Loop Microtissues

CD31 immunohistochemistry staining was performed to identify newly formed vessels in the construct. Newly formed CD31-positive vessels could be detected in all constructs (Figure 7A–D). The fibrin group demonstrated significantly higher vessel number per mm^2^ and proportion of vessel area compared to other groups (Figure 7E,F). No difference was observed among alginate–fibrin, PCL, and control groups.

### 3.7. Infiltration of Immune Cells within the AV Loop Microtissues

To characterize the immune activity in the constructs, CD68, CD163, CD8, CD3, and PD-L1 immunohistochemistry staining was performed. We found sparse to moderate presence of CD68, CD163, and CD8 positive cells in all constructs. As shown in Figure 8, CD68+ and CD163+ macrophage cells could be detected in tumor stroma, connective tissues, and around vessels. Activated CD8+ and CD3+ T-cells infiltrated into tumor regions, stroma, and surrounding vessels mainly in the alginate–fibrin, PCL, and fibrin groups, compared to fewer cells in controls. The distribution of CD8+ and CD3+ T-cells was mostly in clusters, whereas CD68+ macrophages were more diffusely present. Abundant PDL-1 positive cells were detected in all groups except the control group. Although we did not quantify immune cells, it appeared that more positive cells, including CD68+, CD163+, CD8+, and CD3+, were more abundant in the alginate–fibrin group. Importantly, our CD3 antibody was specific for rat T-cells, which demonstrated strong CD3 positive expression (Appendix A).

## 4. Discussion

Engineering 3D tumor models may potentially help to improve drug development [22]. It is critical to develop a relevant and appropriate 3D model to keep a balance between fidelity to human conditions and practical considerations [23,24]. Abundant researchers have demonstrated that the usage of xenograft models is strongly relevant for developing new therapeutics against human cancers [8,25,26,27]. In addition, human breast cancer cell lines like MDA-MB-231 (HTB-26) used in our study have been implemented extensively into preclinical models both in vitro and in vivo as xenografts to investigate progression of breast cancer [2].

The AV loop model was first described by Erol and Spira [28]. This model is suited for engineering vascularized tissues in vivo in an isolated and controlled environment, and can be tailored by varying matrices, cells, and other factors [8]. Moreover, the AV loop model is confirmed to mimic the desired microenvironment more specifically in the chamber than the typical subcutaneous model [9]. In accordance, recently a new vascularized melanoma model was developed using the AV loop by our group with the aid of a novel printable hydrogel consisting of alginate, hyaluronic acid, and gelatin for mimicking the tumor microenvironment [9]. Based on these concepts, we are the first to use this AV loop model for vascularized breast cancer tissue engineering by implantation of human breast cancer cells in combination with different matrices.

Important considerations in scaffold engineering involve pore size, degradability, and biocompatibility with respect to the implant location. Leung et al. demonstrated that tumor cells pre-exposed to a 3D matrix microenvironment in vitro could maintain elevated angiogenic capacity upon implantation in vivo and, as a consequence, lead to enhanced vascularization and accelerated tumor growth [22,29]. For in vivo applications, scaffolds and matrices should not only allow for effective cell seeding and penetration in situ, but should also promote vascularization to create a vascular network for tumor tissue engineering [30]. Thus, it is necessary to identify a suitable cell delivery matrix for maximum cell survival, sustaining a vascular network, tissue infiltration, and tumor formation.

PCL has potential as a synthetic polymer in tissue engineering due to its biomechanical and structural characteristics [31]. In our study, we used PCL scaffolds with a high porosity of approximately 80% with large pore sizes, which allow high oxygen and nutrient transportation throughout the scaffolds. Breast cancer cells HTB-26 constitutively co-expressed keratins and vimentin [32] and oriented along PCL fibers and proliferated in vitro (Figure 2A–D). Fibrin, a minor component of the tumor ECM, may provide an initial structural support for the breast cancer cells [33]. Our work demonstrates that the fibrin concentration affected proliferation of breast cancer cells in a dose-dependent manner (Figure 2E). This observation supported previous findings that indicated that lower fibrinogen concentration leads to a higher proliferation of mesenchymal stem cells in 3D fibrin matrices [34,35]. In addition, soft fibrin gels were shown to promote growth of stem-cell-like cancer cells in 3D culture and enhance their ability to form tumors in mice [36]. Generally, increasing fibrinogen concentrations results in a longer degradation time. However, increasing the fibrinogen concentration to 33 mg/mL could delay the onset of neovascularization until day 12 to 14 after construction of the AV loop compared to the application of 10 mg/mL fibrinogen [28,37]. Thus, in our study, we chose 2% (*w*/*v*) fibrinogen for further implantation because of its higher cell viability and comparatively higher stability (Figure 2E,F). Three-dimensional fibrin gels in combination with other materials, such as PLGA and alginate, enhance the stability of the matrix and provide a complex biomaterial platform to support neovascularization in vitro and in vivo [38,39]. Breast cancer cells were successfully encapsulated in an alginate–fibrin matrix (Figure 2H,I), which is consistent with our previous study [40]. Alginate–fibrin capsules with a diameter of approximately 500 μm used in this study facilitate high cell loading densities, which reduce the number of capsules needed for implantation.

In vivo imaging along with our histological findings were congruent. Notably, the homogeneous distribution of the epithelial marker cytokeratin and the mesenchymal marker vimentin in the alginate–fibrin, PCL, and fibrin groups (Figure 5), confirms the formation of an engineered breast tumor tissue via the AV model. After 8 weeks of implantation, fibrin was found to strongly degrade (Figure 4H). Combined with imaging (Figure 3D,G), the fibrin matrix was presumed to be mainly replaced by connective vascularized tissue and solid tumor tissue. In comparison, the average microtissue size of the alginate–fibrin group was significantly increased (Figure 4D). The high stability over time of the alginate–fibrin matrix represents an advantage compared to the use of fibrin only, as the latter can be more easily degraded enzymatically by tumor cells [36]. Moreover, implementing a radiotracer uptake and imaging in vivo using a PET-CT, the implantation time, and strong cytokeratin-/vimentin staining in the alginate–fibrin matrix determined larger tumors. Thus, we interpret that the alginate–fibrin matrix provided a broad range of chemical cues, principally ECM binding motifs, enabling efficient diffusion of small molecules, which actively drive tumorigenesis [12]. Moreover, alginate–fibrin capsules were placed around the AV loop, enabling bidirectional diffusion of nutrients, oxygen compounds, cytokines, and circulation through their porous structure [41], leading to increased cell proliferation. However, there is a limitation with regard to the in vivo imaging in our study that we could only employ a small number of animals in alginate–fibrin and fibrin groups for imaging. Therefore, we did not analyze statistically and only describe a tendency. We will include more animals in such an analysis in future studies to overcome the disadvantage.

As the degradation period of PCL is approximately 2 years, the functional role of the PCL framework was to enhance the mechanical stability of the cell-laden construct, promoting a healthy biological environment, which could be maintained over a longer time period [42]. In addition, we introduced a 3D printed ring-shaped alginate–fibrin matrix (Figure 2J) around the PCL scaffold. This matrix ring was incorporated in parallel with an AV loop into the implantation chamber, which connected the PCL scaffold with the vascular system. Although small tumor nodules were mainly found between the scaffolds in the PCL group and the connective tissue (Figure 5), the average construct size in the PCL group was significantly smaller compared to the alginate–fibrin group. Perhaps, the phenomenon may be explained by (1) the alginate–fibrin hybrid matrix having more bioactive signals facilitating cell proliferation [12]; and/or (2) the hydrophobicity of PCL scaffolds favoring connective tissue formation in vivo that outgrows and impedes cancer epithelial cell attachment and proliferation, thus affecting tumor development [43].

Ki67 has become a very important predictive and prognostic marker for breast cancer [44]. The in vivo observation, with a 30–40% positive cell rate for Ki67 after implantation of 8 weeks (Figure 6), mimics the primary tumors (often under 50% positive for Ki67) harvested from patients [45]. Abundant proliferative and rare apoptosis cells detected in all groups indicated the biocompatibility of these scaffolds and matrix. Combining the results in Figure 7 and Figure 8, CD31, CD68, CD163, and CD8 -positive cells, which were believed to be endothelial cells, macrophages, and immune cells, were observed in all groups. These vast amounts of Ki67-positive cells observed in all groups include not only implanted tumor cells but also infiltrating immune cells and stromal cells, such as fibroblasts stained by vimentin.

A higher/faster tumor growth in vivo could be based on a higher degree of neovascularization within the AV loop chamber. The composite constructs were successfully vascularized, as demonstrated by 3D-imaging and histology. The newly grown network was characterized by positive CD31 staining (Figure 7). The significantly higher density and area of vessels in the fibrin group compared to the other groups confirmed the excellent ability of fibrin to promote angiogenesis, which is consistent with a previous study [46]. Combined with the finding of the remarkably decreased size of microtissues in the fibrin group, the high number of vessels in the fibrin group indicated that sprouting vessels are tightly packed in a small volume of hydrogels, which may be attributed to the fast degradation rate of fibrin. Notably, abundant vessels could be visualized in the control group where no tumor cells were implanted, suggesting the venous and interpositional venous graft segments may play a vital role in giving rise to many capillaries and large vessels. As the engineered breast tumor was established in an isolated chamber, we propose that the model could probably also be expanded to study tumor metastasis via the newly formed vasculature.

Referring to the size and structure, breast tumors are heterogeneous. Crosstalk between cancer cells and cancer-associated cells, such as resident mesenchymal support cells, endothelial cells, and inflammatory immune cells, fuels and shapes tumor development [47]. Infiltration of macrophages in the microtissues (Figure 8), detected by CD68 and CD163 immunohistochemistry, may be positively associated with a high level of angiogenesis and tumor progression to malignancy [48,49]. Interestingly, CD8 and CD3 positive immune cells are also distributed evenly in all the microtissues, indicating some sort of immune response. Although the athymic nude rats used in this study are always largely deficient in mature T cells, nude rats could develop T-like cells expressing CD3, CD8, and the T-cell receptor (TCR) with increasing age, especially after 4–6 months [50]. Despite that these T-like cells resemble the phenotype of T-cells found in normal rats, they lack alloreactivity in vivo, and their TCR repertoire is more of an oligoclonal nature [51]. Programmed death ligand 1 (PDL1), widely expressed on the surface of dendritic cells, B-cells, and multiple tumor cells, including breast cancer, has been associated with large tumor size, high grade, high proliferation, HER-2-positive status, leading to a poorer clinical outcome [52,53]. Higher expression of the PDL1 in tumor cells allows tumors to evade the host’s immune system, leading to a worse clinical outcome [54]. These exciting findings provide strong evidence that the designed breast cancer model mimics human cancer development involving a complex tumor–microenvironment interaction.

## 5. Conclusions

In summary, this study describes a novel breast tumor model for the first time in nude rats with the aid of 3D printing to customize a scaffold or matrix that could provide an “on-demand” profile for cell growth and proliferation leading to engineered tumor formation. Although fibrin showed an excellent ability to promote angiogenesis, its rapid degradation limited its application. The alginate–fibrin hybrid matrix may be considered as the most attractive for tumor engineering within the AV loop. As we were able to establish an isolated tumor microenvironment within this study, this unique model could be used as a platform to investigate tumor–stroma interactions and explore breast cancer metastasis through the newly formed vasculature. In the future, a personalized AV loop model could be established by transplantation of breast cancer patient-derived tumors for drug-testing purposes to customize the individual therapy. This newly developed tumor model may find broad utility in culturing of tumors in vitro and in vivo to aid in the investigation of cancer progression and as a new testing system for anti-cancer therapies.

## Figures and Tables

**Figure 1 bioengineering-09-00280-f001:**
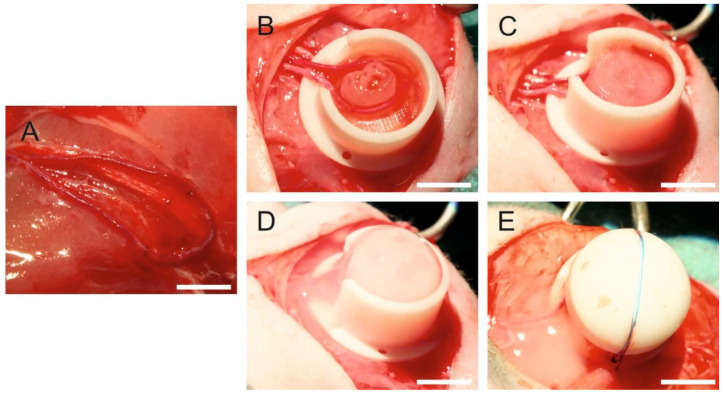
Arteriovenous (AV) loop operation in the rat. (**A**) The vessels were separated and the vein graft harvested from the right side was anastomosed with the femoral vein and artery of the left side to form the AV loop. (**B**) One 3D alginate–fibrin ring with cells was placed inside the loop, which was on three PCL scaffolds. (**C**) The other ring with cells was put around the AV loop, and another three PCL scaffolds with HTB-26 cells were put on the AV loop. (**D**) One piece of 3D printed alginate–fibrin sheet without cells was placed on the top and the chamber was filled with matrix. (**E**) The chamber was closed with a lid and fixed on the thigh. Scale bar 1 cm.

**Figure 2 bioengineering-09-00280-f002:**
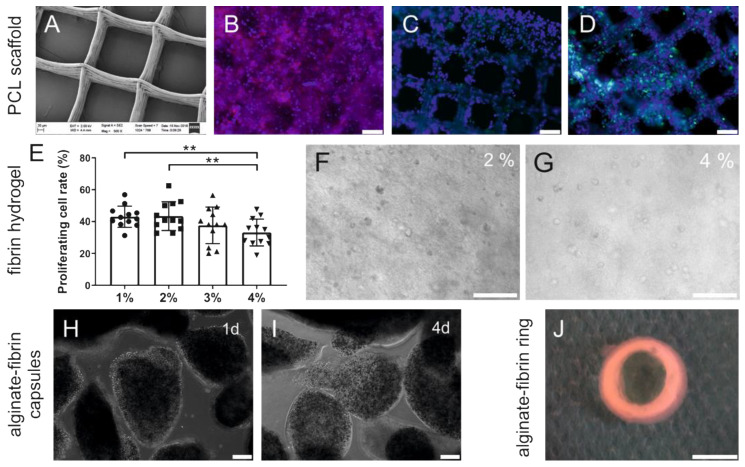
Cell morphology, attachment, distribution, and proliferation in PCL scaffolds, fibrin, and alginate–fibrin hydrogel. (**A**) Morphological structure of PCL scaffold using SEM. (**B**) Immunofluorescence for wheat germ agglutinin (red) combined with Hoechst33342 counterstaining (blue) to visualize the nuclei on HTB-26 cells seeded on PCL scaffolds after 7 d culture. Scale bar 100 μm. (**C**,**D**) Representative images of HTB-26 seeded on PCL scaffolds with cytokeratin immunofluorescence (green) (**C**) and cell proliferation detected using Ki67-immunofluorescence (green) after 3 d culture (**D**). Counterstaining with DAPI (blue) to visualize the nuclei. Scale bar 100 μm. (**E**) Effect of different fibrin concentrations on cell proliferation determined by Ki67 immunofluorescence at 18 d. Scale bar 100 μm. (**F**,**G**) Morphology and distribution of HTB-26 cells cultured in 2% and 4% fibrin hydrogels at 4 d. Scale bar 100 μm. (**H**,**I**) Representative images of HTB-26 cells encapsulated in alginate–fibrin capsules at 1 d (**H**) and 4 d (**I**). Scale bar 100 μm. (**J**) Macroscopic structure of the 3D-printed alginate–fibrin ring. Scale bar 10 mm. ** *p* <0.01.

**Figure 3 bioengineering-09-00280-f003:**
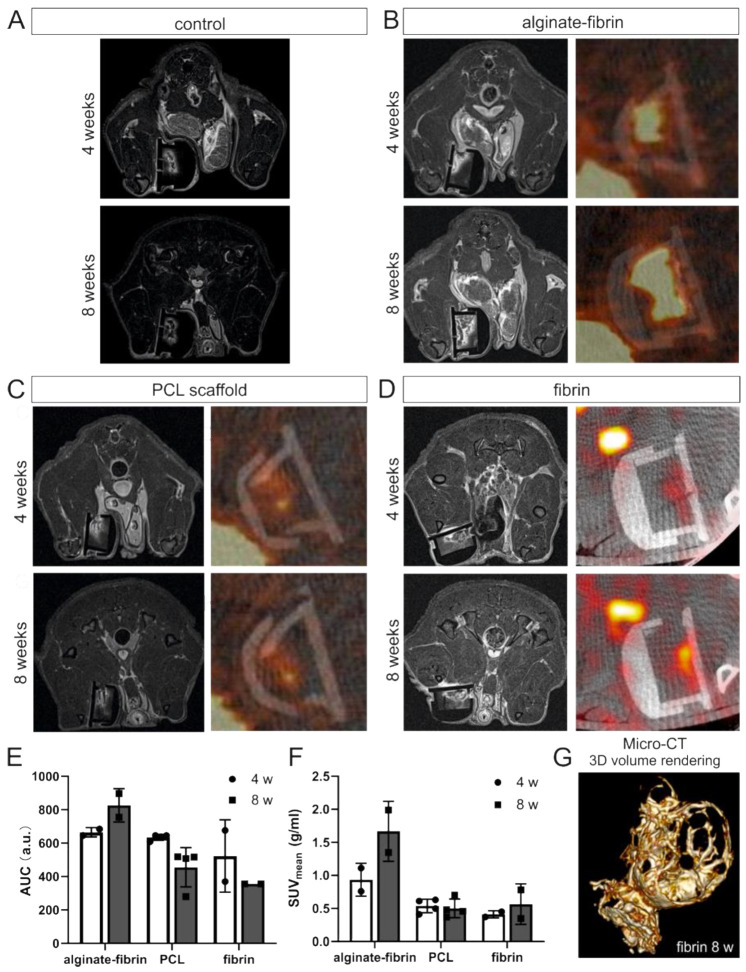
Imaging analysis. Representative MRI and PET-CT imaging in control (**A**), alginate–fibrin (**B**), PCL (**C**), and fibrin (**D**) groups. Yellow arrows indicate the perfused area within the implantation chamber in MRI imaging. White arrows indicate the tracer-avid area formed by tumor in PET-CT imaging. One representative image per group and per time-point is shown. Quantitative analysis of AUC (measured in a.u. arbitrary units) (**E**) and SUV_mean_ (**F**) derived from PET-CT. (**G**) Representative 3D micro-CT images of the microvasculature in fibrin group after perfusion with an intravascular contrast agent (yellow perfused vessels) after 8 weeks implantation.

**Figure 4 bioengineering-09-00280-f004:**
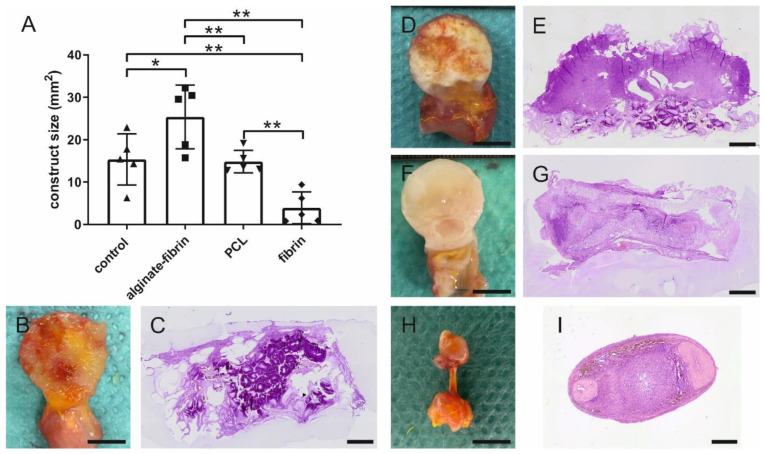
Morphometric analysis of the construct size. (**A**) Quantitative analysis of the construct size of H&E stained cross sections. Representative images of H&E staining of cross sections and macroscopic pictures of the constructs derived from control (**B**,**C**), alginate–fibrin (**D**,**E**), PCL (**F**,**G**), and fibrin (**H**,**I**) groups explanted after 8 weeks. Scale bar (**B**,**D**,**F**,**H**): 1 cm, (**C**,**E**,**G**,**I**): 1 mm. * *p* < 0.05, ** *p* < 0.01.

**Figure 5 bioengineering-09-00280-f005:**
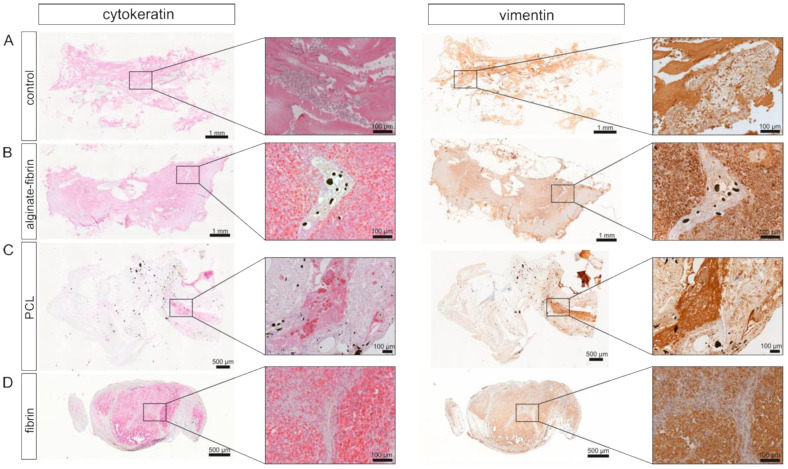
Immunohistochemical staining for cytokeratin and vimentin in the constructs. Representative images of cytokeratin and vimentin immunohistochemical staining in control (**A**), alginate–fibrin (**B**), PCL (**C**), and fibrin (**D**) groups presented at low magnification (40×) and high magnification (200×). Black arrows indicate the cytokeratin-positive cells. Yellow arrows indicate the vimentin-positive cells.

**Figure 6 bioengineering-09-00280-f006:**
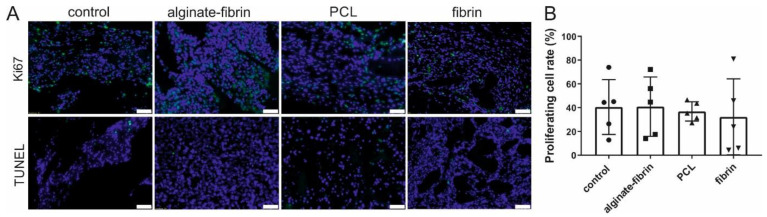
Proliferation and apoptosis in the constructs. (**A**) Representative images of Ki67 and TUNEL staining to investigate cell proliferation and apoptosis, respectively. (**B**) Quantification of the proliferation rate in different constructs. Scale bar 100 μm.

**Figure 7 bioengineering-09-00280-f007:**
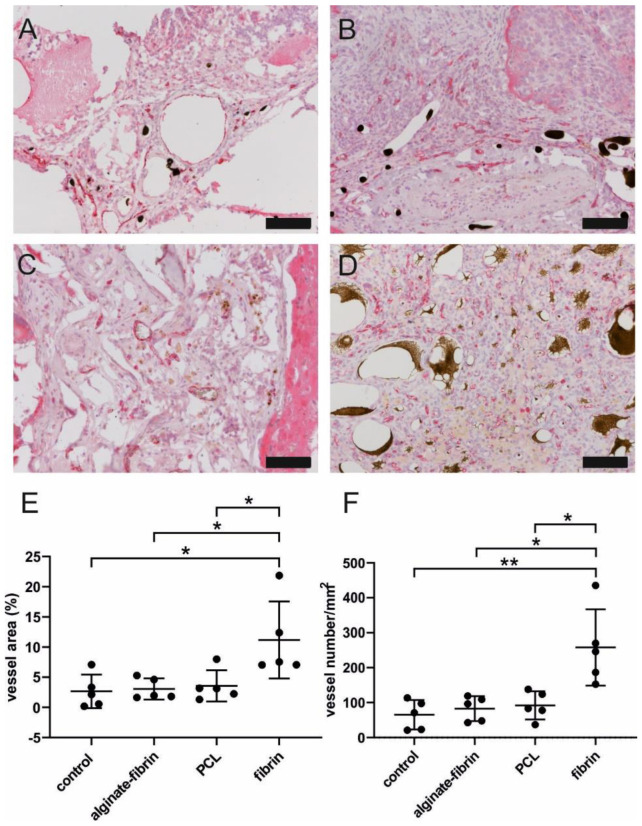
Vascularization of constructs. Representative microscopic images of CD31-immunohistochemistry staining of control (**A**), alginate–fibrin (**B**), PCL (**C**), and fibrin (**D**) groups. Scale bar 100 μm. Quantification of area per vessel (**E**) and vessel number per ROI (**F**) by CD31 staining. * *p* < 0.05, ** *p* < 0.01.

**Figure 8 bioengineering-09-00280-f008:**
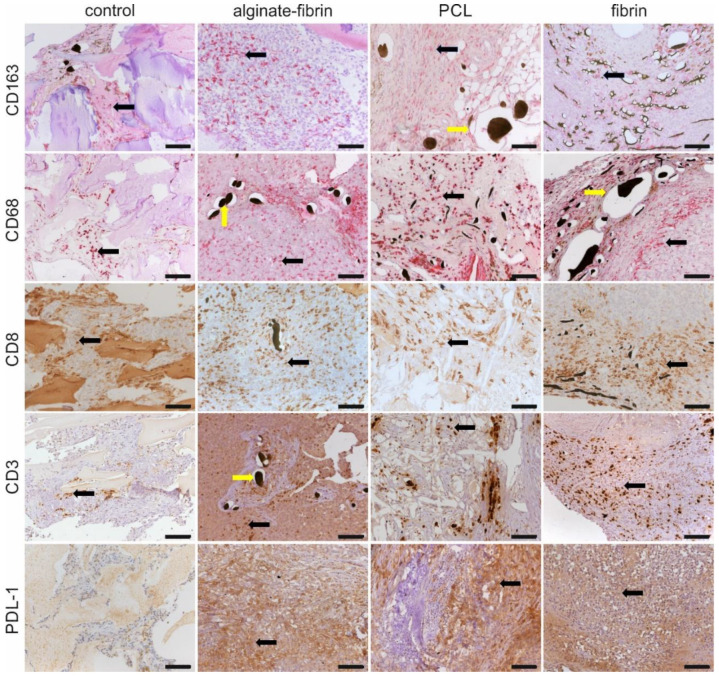
Immunohistochemical staining for tumor associated macrophages and immune cells in the constructs. Representative images of CD163, CD68, CD8, CD3, and PDL-1 immunohistochemical staining in control, alginate–fibrin, PCL, and fibrin groups. Scale bar 100 μm. Black arrows indicate positively stained cells. Yellow arrows indicate perfused vessel.

## Data Availability

The data presented in this study are available on request from the corresponding author.

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
