# Peer review of "An Innovative Arteriovenous (AV) Loop Breast Cancer Model Tailored for Cancer Research"

_bioengineering, 2022, doi:10.3390/bioengineering9070280_

Round 1

Reviewer 1 Report

An Innovative Arteriovenous (AV) Loop Breast Cancer Model Tailored for Cancer Research by An et al. aimed to develop an advanced human breast cancer model by using the AV loop rat model and implantation of breast carcinoma cells (HTB-26) in different scaffolds (polycaprolactone – PCL, fibrin, and alginate-fibrin). They performed comprehensive analyses in vitro and in vivo.

Cell proliferation and rare apoptotic cells were detected in all groups as well as vascularization. Significantly higher density and area of vessels were obtained in the fibrin group compared to the other groups confirming the high ability of fibrin to promote angiogenesis.

The authors provided some important findings and recommended the alginate-fibrin matrix as advantageous over PCL and fibrin only. Larger tumors with high proliferative potential were developed in alginate-fibrin and it was more stable and resistant to enzyme degradation.

Moreover, alginate-fibrin in the AV loop provided extracellular matrix binding motifs and enabled efficient diffusion of small molecules (bidirectional diffusion of nutrients, oxygen compounds, cytokines, and circulation through their porous structure).

The authors’ findings provide evidence that their breast cancer model can mimic human cancer development involving a complex tumor-microenvironment interaction and imply its usage to study tumor metastasis via the newly formed vasculature.

Having all these in mind, I warmly recommend this work for publication in Bioengineering (MDPI).

Author Response

Thank you very much for the review. One of the co-author, a native English speaker, has helped to checked and improve the language.

Reviewer 2 Report

This is an interesting study. The 3-dimensional in vivo arteriovenous loop model provides a good platform to study tumour growth, proliferation and angiogenesis and could be used to study the effects of drugs on these properties in a live 3D context. However, this would not be high throughput and would require considerable surgical skill to apply the constructs which would reduce the practicality of this approach as a generalised model. Also some of the images are not clearly explained and this makes it hard to appreciate what the authors surmise form the data shown.

I have some comments/suggestions and queries for the authors as follows:

1. It appears that the cancer cells are immobilized in these gels? If so is the pathophysiological relevance of these matrices for tumour progression diminished as migration is not possible?

 2. Schematic highlighting the different construct compositions and final number of cells implanted would help (at least as suppl. Material).

3. Line 197: “For implantation 5 × 106 HTB-26 were diluted in 100 μl thrombin”. Is this correct? Or was it 5x106 HTB-26/ml

4. Line 330: Isn’t KI-67 a marker of proliferation and not viability (at least not directly)??

5. Line 420 -424: the legend needs more detail to explain the results in Figure 3. Also arrows highlighting key features in each image would help.

6. Line 460-463: again this legend and the images need more detail. What is the positive staining in each case. To me it looks like there is positive vimentin staining in the control in which no breast cancer cells were loaded. Same suggestion and query around the controls for Figure 6. Why are there Ki67 positive cells in the control group if they were not implanted with cells? Are these infiltrating immune cells or other stromal cells? Authors should comment more on this.

7. Figure 8: indicate with arrows examples of positive cells, and indicate in tumor vs stroma/connective tissues and vessels clearly in images. Can this data be quantified in any way?

8. Line 542-544: the connection to the previously presented data is missing.

Reviewer 3 Report

The authors should revise their conclusion after addressing the points below:

- What does the blue staining correspond to in Figure 2B and 2D? Please, indicate what each channel corresponds two in double stained samples.

- Figure 2F and 2G: no difference in colony size is apparent in the images shown here.

- Figure 2H: capsules do not appear spherical unlike indicated in the main text. 

- Figure 2J: what does the red colour corresponds to?

- Figure 3 and F: only two measurements\animals per time-point are analysed in the case of aliginate/fibrin or fibrin, which limits the strength of the final results

-Figure 4: would it be also possible to quantify the number of tumour cell nuclei per explant? It looks like that despite their size PCL-constructs contain less cells than fibrin-constructs

Round 2

Reviewer 3 Report

The authors addressed most of the comments raised by the reviewers.

This manuscript is a resubmission of an earlier submission. The following is a list of the peer review reports and author responses from that submission.